


# Full scale experiments to examine the role of deadwood on rockfall dynamics in forests

Adrian Ringenbach[1,3], Elia Stihl[1], Yves Bühler[1], Peter Bebi[1], Perry Bartelt[1], Andreas Rigling[2,3], Marc Christen[1], Guang Lu[1], Andreas Stoffel[1], Martin Kistler[1], Sandro Degonda[1], Kevin Simmler[1], Daniel Mader[3], and Andrin Caviezel[1]

[1]WSL Institute for Snow and Avalanche Research SLF, 7260 Davos Dorf, Switzerland
[2]Swiss Federal Institute for Forest, Snow and Landscape Research WSL, Zürcherstrasse 111 8903 Birmensdorf, Switzerland
[3]Department of Environmental Systems Science, Institute of Terrestrial Ecosystems, ETH Zürich, Zurich, Switzerland

**Correspondence:** Adrian Ringenbach (adrian.ringenbach@slf.ch)

**Abstract.** Forests are rockfall-protective ecological infrastructures, as a significant amount of kinetic energy is absorbed during consecutive rock-tree impacts. Although many recent works have considered rock impacts with standing trees, the effect of lying deadwood in forests has not yet been considered thoroughly, either experimentally or numerically. Here, we present a complete examination of induced rockfall experiments on a forested area in three different management stages. The trilogy is

conducted in a spruce forest stand (i) in its original state, (ii) after a logging operation with fresh, lying deadwood and (iii) after the removal of the deadwood. The tests allow us to directly quantify the effect of fresh deadwood on overall rockfall risk for the same forest (slope, species) under three different conditions. The study yields quantitative results on the barrier efficiency of the deadwood logs as only 3.6 % of the rocks surpass the deadwood section. The mean runout distance is reduced by 42 %. Conversely, the runout distance increases by 17 % when the cleared stand is compared to the original forest. These

results quantitatively confirm the benefits of nature-based mitigation measures integrated into forestry practice and we show how modern rockfall codes can be extended to incorporate such complex, but realistic forest boundary conditions.

## 1 Introduction

Rockfall is a common natural hazard in mountain regions. Up to certain rockfall energies, protection forests are effective ecological infrastructures to reduce rockfall intensities and thus the damage on vulnerable facilities as falling rocks are decelerated

by consecutive tree impacts. This has been shown by real scale experiments at the slope scale in a mixed Abies-Picea-Fagus forest (Dorren and Berger, 2005) as well as on single trees of different species including *Picea abies*, (Lundström et al., 2009) and *Ailanthus altissima*, (Wunder et al., 2018). Implementation of these findings into three dimensional rockfall models, e.g. Rammer et al. (2010); Dorren (2012); Toe et al. (2017); Lu et al. (2020), facilitated additional investigations to quantify the protective capacity of mountain forests using numerical tools at the forest stand (Stoffel et al., 2006; Woltjer et al., 2008; Moos

et al., 2017) and regional scales (Dupire et al., 2016; Lanfranconi et al., 2020).

Experimental tests to investigate the role of deadwood for rockfall mitigation have also been performed at the laboratory (Ammann, 2006; Olmedo et al., 2015) and slope (Bourrier et al., 2012) scales. The laboratory studies provided the first quan-


titative insights into the protective effects of deadwood but do not accurately represent natural deadwood configurations after disturbances since the investigators used clumped *Picea abies* and *Fagus sylvatica* specimens with relatively small mean diameters of 26 cm and 6 cm (Ammann, 2006; Olmedo et al., 2015). The field studies of Bourrier et al. (2012) used larger 63 cm diameter logs fixed to tree stumps using steel cables. How deadwood is naturally fixed (jammed) between trees and the ground is essential to understand its protective capacity and therefore man-made fixations may also not represent natural conditions in mountain forests. As a consequence, present approaches to include deadwood into a three dimensional rockfall model are based on slope or roughness adaption methods combined with higher, and empirically determined, rock-ground restitution coefficients (Fuhr et al., 2015; Costa et al., 2021).

Neither the different experimental nor the modeling approaches take piled stems into account, which are often the result of overturned trees during windthrow events. Beside windthrows (Feser et al., 2015), also bark beetle outbreaks (Jönsson et al., 2009) and forest fires (Mozny et al., 2021; Jain et al., 2020) are likely to increase in frequency or amplitude due to climate change and land-use legacies. Allowing natural processes without salvage logging after windthrow and without sanitary felling after bark beetle are increasingly promoted as adequate management option (Kulakowski et al., 2017; Sommerfeld et al., 2021). In forests with a protection effect against rockfall hazards it is decisive to know more on the short- and longterm effects of piled stems on rockfalls after natural disturbances or management interventions.

In this paper, we present the results of three induced rockfall experiments within the same mountain forest but in three different management states. First, we performed rockfall experiments in the original, undisturbed forest. In the next series of tests, the effect of lying, partly piled, fresh deadwood in the upper third of the slope was tested. The "deadwood" was created after a forest management intervention and was therefore in a fresh condition with maximal physical resistance which diminishes with increasing wood decay. In the final test series, the deadwood was cleared (the final forest was therefore sparser than the original forest). All three tests were therefore performed in the same spruce stand to explore the protection effect of lying deadwood quantitatively. To obtain a physical understanding of how deadwood functions as a mitigation measure, a three-dimensional rockfall model - using truncated cones as deadwood-logs - was applied to model the experiments.

## 2 Material and methods

### 2.1 Study site: Surava

The study site (46.65720° N, 9.60497° E, 1120 m a.s.l.) covers 0.54 ha of north-west exposed, roughly 35° steep mountain forest in the community Surava within the municipal area of Albula/Alvra, Switzerland. In a complete forest inventory the diameters at breast height (DBH), tree species and GNSS-positions of all trees with DBH $\geq$ 8 cm were recorded, resulting in a total of 462 trees, a stand density of 855 trees ha$^{-1}$ and a mean DBH of $23.9 \pm 11.7$ cm, see Figure 1.a). The principal tree species are Norway spruce (78.8 %, *Picea abies* (L.) Karst.), European larch (9.5 %, *Larix decidua*), Mountain pine (6.7%, *Pinus mugo*) and Silver fir (4.1 %, *Abies alba*). Further single Rowans (*Sorbus aucuparia* L.), Beeches (*Fagus sylvatica*) and Whitebeams (*Sorbus aria*) were registered.



## 2.2 Experimental set-up

Rockfall experiments were conducted during three different forest states: in the original forest (**ORG**), after a logging job with lying, partly piled deadwood (**DW**) and subsequently with this area cleared (**CLR**). The logging-job area was located in the top third of the slope, covered about 700 m² and comprised 53 trees, of which 26 trunks were left lying for the CLR-experiments (Fig. 1.a). We used perfectly symmetrical reinforced *EOTA* concrete rocks with a mass of 45 kg and the defined geometry according the ETAG 027 (2013) guideline (Fig. 3.d). This mass allowed still a manual rock handling, even if rocks stuck between deadwood logs. In a borehole through its center of mass, in situ StoneNode v1.3 sensors were mounted to record rotational velocities ($\omega$) in all three axis up to 69.8 rad·s$^{-1}$ (= 4000 °·s$^{-1}$) and accelerations up to 3922.66 m·s$^{-2}$ (= 400 $g$) with a sampling rate of 1 kHz (Niklaus et al., 13.03.2017 - 15.03.2017; Caviezel et al., 2018).

The rocks were repeatedly released manually from the same starting position at 1118.5 m a.s.l.: 42 runs in the **ORG**, 28 runs in the **DW**, and 41 runs in the **CLR** state, of which at least 73 % per state include the in situ sensor streams. The run-number difference between the forest states resulted due to time constraints of the forest service, as just one field day was possible for the **DW**-state, but two field days were used for the other states. The deposition points of the rocks were measured in with a high-precision Trimble GeoXH differential handheld GNSS. The mean achieved horizontal accuracy in this steep north-west exposed forest was 1.6 m. Thereafter the rocks were winched back from its deposition point to the forest road and from there transported by four wheel motorcycle back to the release-point.

To compare the deposition pattern between the three forest states quantitatively, the geographic mean centre (GMC), the third standard deviation ellipse (SDE) and its radius of the long (SDE$_{la}$) and short axis (SDE$_{sa}$) were calculated. The mean run out distance (MROD) is the euclidean, slope parallel distance between release point and GMC.

## 2.3 Sensor data processing

In order to detect frontal impacts (FI) on opposing objects such as standing trees, overturned root plates or larger rocks, we applied gyroscopic data-analysis, rather than evaluate the acceleration data, since the latter vary strongly depending on the incoming translation velocity and especially the material being hit. Regardless of the material, after an FI, a sharp reduction in the resulting rotational velocity will occur. The *Matlab* function *ischange* (Killick et al., 2012) with its linear method is applied to find abrupt slope changes on the smoothed resultant gyroscopic data stream. To reduce the rare artifact spikes in the rotational data stream, 5 different moving window sizes (0.001 s, 0.021 s, 0.061 s, 0.101 s and 0.151 s) with their corresponding thresholds are applied, while at least 3 of 5 thresholds must be exceeded for an FI to be counted. The used slope limits are $\leq -100$, $\leq -29$, $\leq -21$, $\leq -9.75$ and $\leq -6.5$.

## 2.4 Rockfall simulations

The experimental rockfall trilogy was reproduced through simulations with a rigid body rockfall code (Leine et al., 2014, 2021) under consideration of compactable soils (Lu et al., 2019) and of single standing trees (Lu et al., 2020). Based on the measured DBH, the in RAMMS::ROCKFALL used tree height was estimated based on $H = DBH^{\frac{1}{1.25}}$ (e.g. Dorren (2017)). To represent



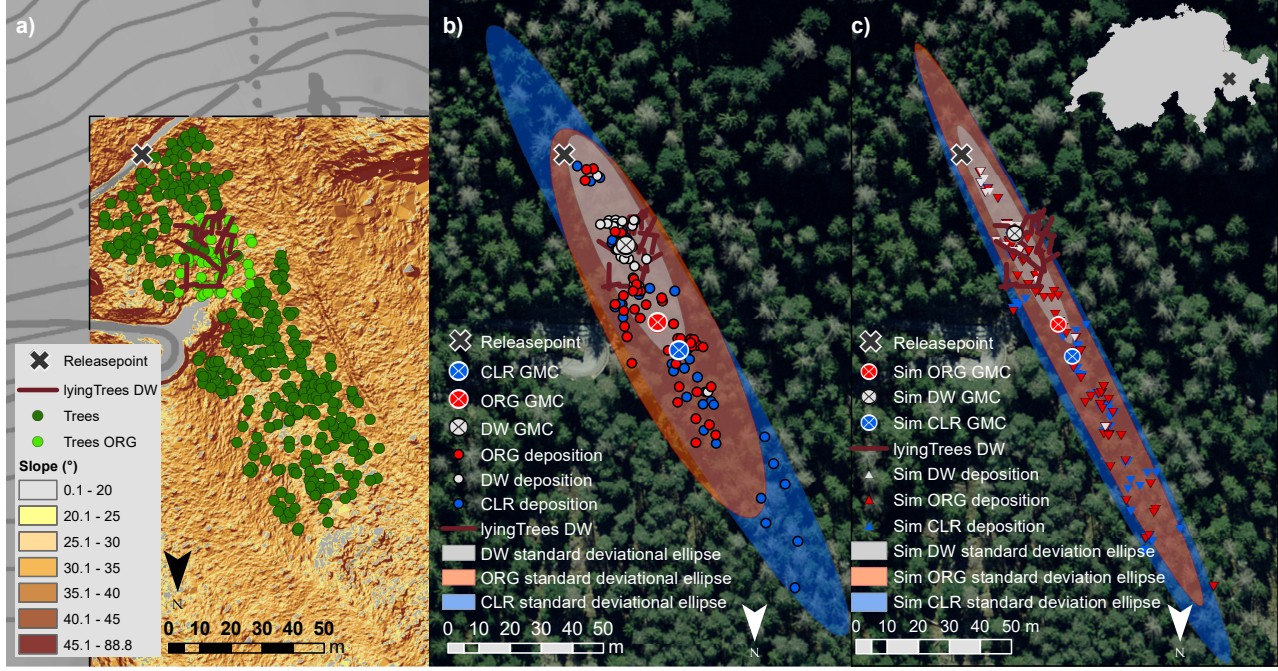

**Figure 1.** Overview of the experimental test site Surava: a) Slope map with the standing trees during the whole trilogy (dark green circles), during ORG state (light green circles) and lying deadwood during DW (brown lines). b) Deposition points of all 111 runs of the 45 kg $EOTA_{111}$ rocks including all three forest states: ORG in red, fresh DW in grey and CLR in blue. The corresponding geographic mean centers (GMC) are shown with crossed symbols in the corresponding color. Additionally the third standard deviation ellipse (SDE) of each data set, are depicted as transparent ellipses. c) deposition point of the total 300 simulated trajectories under consideration of the three forest states. The GMC and SDE are in the corresponding colors of the experimental results. Source of topographical map (a) and orthophoto (b,c): Federal Office of Topography

the conditions of the **DW** state, we introduce additionally a new deadwood module, which includes lying deadwood as rigid, three-dimensional cones. During fieldwork, all 26 trunk GNSS positions were recorded; we took for all deadwood-logs a maximal diameter of 40 cm. If logs were lying on top of each other, we considered the first mentioned in the GNSS-file as the lower log.

Current surface models from the Swiss Federal Office of Topography swisstopo are available with resolutions as fine as 0.5 m grid size. The need to account for mesoscale roughness effects as well as customized ground point detection algorithms, nonetheless demanded a site-specific airborne LiDAR mission. The LiDAR point cloud has a density $\geq 500$ points$\cdot$m$^{-2}$ and was scanned by an Trimble AC60 sensor, after the CLR-experiments (Fig. 2). The generated digital surface models with a resolution of 0.05 m, processed in the LAStool-framework with the $-extrafine$ option, specialized to detect ground points in steep mountainous regions (Isenburg, 2021), was used for the simulation of all forest states.




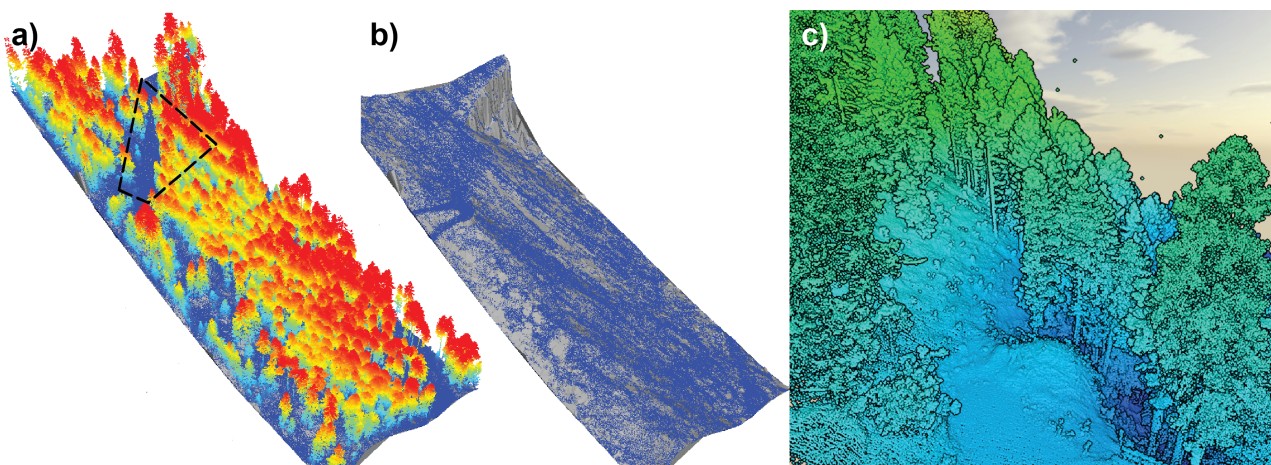

**Figure 2.** a) The acquired LiDAR point cloud colored according to the height above ground (blue = 0 m, red $\geq$ 23 m). b) LAStool post-processed ground points used to generate a high-resolution digital surface model for the rockfall simulations. c) Zoomed-in section of the cleared area as indicated in a). The mesoscale surface roughness is clearly discernible in the ground points in the cleared area as well below the remaining tree canopy.

The soil mechanical strength $M_e$ and the scar drag coefficient $C_d$ are the soil parameters to calibrate (Lu et al., 2019; Caviezel et al., 2019b). We performed simulations sweeps within the relatively wide range of $0.8 \leq M_e \leq 2.6 \, \mathrm{kN \cdot m^{-2}}$ and $0.7 \leq C_d \leq 2.0$ and 300 orientations to narrow down the used parameter ranges for the in-depth calibration. Subsequently a model run with 3000 initial rock orientations per forest state and $M_e$ and $C_d$ pairs within the ranges of $1.6 \leq M_e \leq 2.1 \, \mathrm{kN \cdot m^{-2}}$ and $1.6 \leq C_d \leq 2.1$ and with a step size of 0.1 was performed. The calibration is based on the comparison of simulated versus experimental deposition pattern. The fit per state was calculated and summed up in the overall fit (OF) over all states:

$$S_{\mathrm{FIT}} = |\mathrm{MROD} - \mathrm{MROD}_{sim}| + |\mathrm{SDE}_{\mathrm{la}} - \mathrm{SDE}_{\mathrm{lasim}}| + |\mathrm{SDE}_{\mathrm{sa}} - \mathrm{SDE}_{\mathrm{sasim}}| \tag{1}$$

where the scenario $S$ is either **ORG**, **DW**, or **CLR**. The sum of all the scenario fits amount to the overall fit

$$\mathrm{OF} = \mathrm{ORG}_{\mathrm{FIT}} + \mathrm{DW}_{\mathrm{FIT}} + \mathrm{CLR}_{\mathrm{FIT}} \tag{2}$$

To reduce scaling effects due to the massively higher number of simulations relative to the experiments, additionally packages of 100 trajectories were randomly drawn and compared to the results of the experiments.



**Table 1.** Resulting statistics of the experiments (left) and Simulations (right) for the original forest state (**ORG**), the state with lying, fresh deadwood (**DW**) and the deadwood cleared area (**CLR**)).

| | | Experiments | | | Simulations | | |
|---|---|---|---|---|---|---|---|
| | | **ORG** | **DW** | **CLR** | **ORG** | **DW** | **CLR** |
| N° Runs/with Sensor data | | 42/42 | 28/21 | 41/34 | 100/100 | 100/100 | 100/100 |
| mean run out distance± SD | (m) | 74.3±27.2 | 43.1±15.4 | 87.5±42.7 | 75.4±39.8 | 37.2±15.7 | 88.4±43.4 |
| mean shadow angle± SD | (°) | 34.4±1.2 | 35.6±0.8 | 34.2±1.3 | 34.9±1.3 | 36.2±0.8 | 34.6±1.7 |
| mean run time ± SD | (s) | 19.2±6.4 | 13.7±5.2 | 21.2±11.4 | 27.7±16.2 | 13.1±6.0 | 31.5±16.6 |
| mean max. acc. per run ± SD | ($g$) | 265.8±98.1 | 217.0±83.0 | 265.3±85.4 | 152.6±49.4 | 138.5±47.2 | 164.3±45.7 |
| mean max. $\omega$ per run ± SD | ($rad \cdot s^{-1}$) | 48.3±6.5 | 45.6±6.3 | 45.7±10.5 | 30.7±3.6 | 30.7±3.7 | 31.1±3.1 |
| mean average $\omega$ per run ± SD | ($rad \cdot s^{-1}$) | 20.8±3.3 | 17.3±2.8 | 18.7±4.5 | 14.2±2.0 | 14.6±2.2 | 14.4±2.0 |
| mean N° $FI$ per run ± SD | (-) | 0.88±0.89 | 0.43±0.51 | 0.57±0.65 | - | - | - |
| mean N° $FI$ per run time ± SD | ($s^{-1}$) | 0.049±0.049 | 0.032±0.041 | 0.031±0.043 | - | - | - |
| mean velocity ± SD | ($ms^{-1}$) | 3.7±1.2 | 2.9±0.6 | 3.7±0.8 | 3.0±0.4 | 3.1±0.5 | 3.0±0.4 |

## 3 Results

### 3.1 Rockfall experiments

None of the 111 released EOTA$_{111}$ rocks were capable of breaking a tree or a deadwood log. The effect of lying deadwood had an important influence on the observed rockfall dynamics for the investigated weight class summarized in Tab. 1 and visualized in Fig. 1.b: The mean slope parallel run out distance (MROD) is reduced by 42.1% when comparing the **DW** to the **ORG** state. The effect of the removed standing trees between **ORG** and **CLR** is visible in the prolongation of the MROD by 17.7%. The DW$_{MROD}$ reduction is accompanied by an increase of the mean shadow angle DW$_\alpha$ by 1.3° and 1.4° compared to ORG$_\alpha$

and CLR$_\alpha$. The similar grouping among the forest configurations is also for the in situ measured data visible: The two-sample t-test produced significant ($\alpha$ = 0.05) differences between the mean run time, the maximal accelerations of the **DW** state and the corresponding variables of the **ORG** and **CLR** states. Within **ORG** and **CLR**, the differences are not significant within the chosen level.

The number of detected FI on opposing objects was between 0.90 (**ORG**) and 0.43 (**DW**) per run. Based on slope parallel

distance between starting and deposition point and the run time from the sensor stream, the mean-velocity for each run is calculated. The longer MROD influences also the mean run time, but not linearly: rocks in the **DW** state were moving on average for 14 s, which results in a mean velocity of 2.9 m s$^{-1}$. The states **ORG** and **CLR** feature run times of 19 s and 23 s, respectively. In conjunction with their longer run out distances the surpass the **DW** state by a 28% higher velocity.



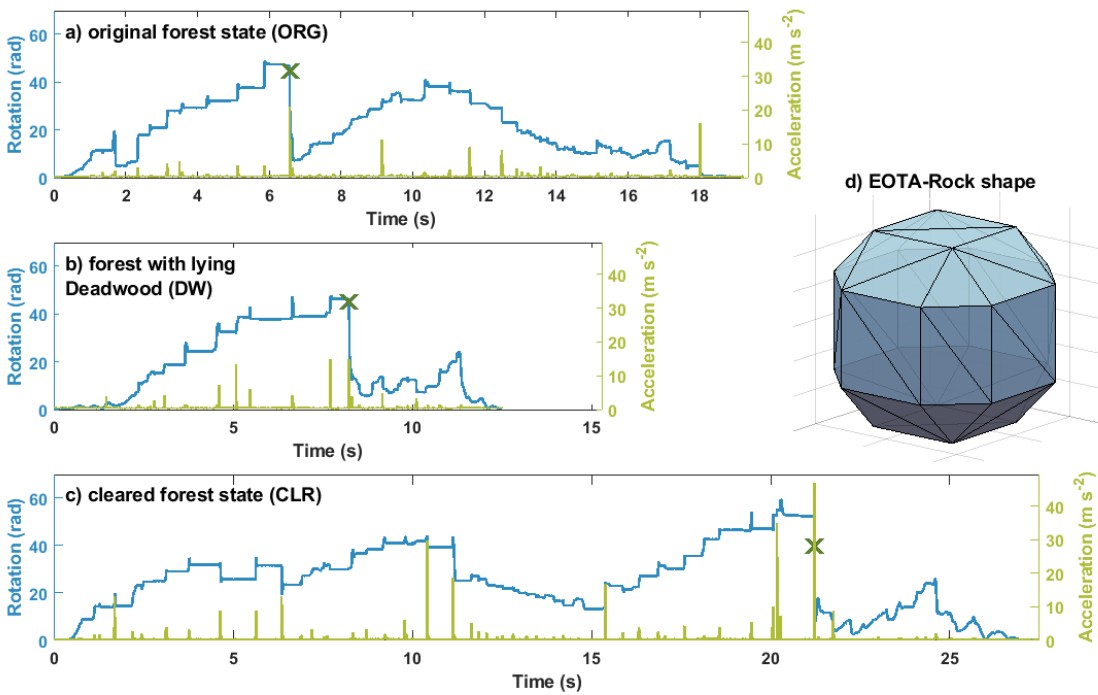

**Figure 3.** Example of three typical StoneNode data streams (a-c) measured within the 45 kg EOTA$_{111}$ rocks (d): in blue the resulting rotational velocities, in green the resulting impact accelerations of a) an ORG-run, b) a DW-run and c) a run within the CLR-stand. The detected frontal impacts are marked as dark green crosses.

## 3.2 Rockfall Simulations

The parameter pair M$_e$ = 2.0 and C$_d$ = 1.9 yield the best overall fit over all 3000 simulation (OF = 83.0 m) and of any randomly sampled package of 100 trajectories (OF = 58.7 m). The following analysis (Tab. 1 and Fig. 4) focuses on this package. The experimentally observed MROD reduction of the deadwood was recognizable in the simulations: The **DW**-simulations had a slightly (5.9 m, 14%) shorter MROD as the experiments, but a similar length of the SDE$_{la}$. While the MROD and its SDE$_{la}$ of the **CLR** state-simulations were both within the experimental GNSS-accuracy, the ORG simulations show a

good fit in the MROD, but a larger longitudinal spread of SDE$_{la}$ as recorded during the experiments. The overall good fit of the simulated MROD consequently leads to a good agreement of the simulated mean shadow angles. The simulated mean run time for the shorter runs of the **DW** state are consistent with the experiments (-4.4%). The run times for the **ORG** (+45.8%) and **CLR** (+37.6%) are by contrast considerably overestimated and the velocities therefore underestimated.

Although the trend of the simulated average maximum accelerations between the forest states is correct, the simulations

globally underestimate the experimental values by roughly 40%. The simulated maximal rotational velocities per run are

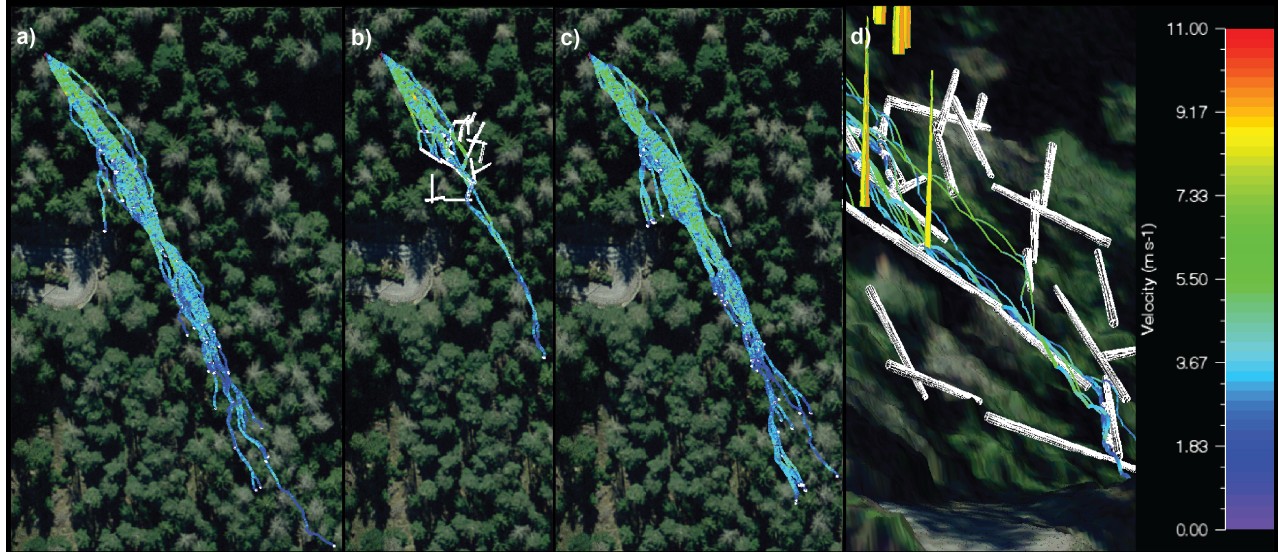

**Figure 4.** Velocity comparison of the 100 simulated rockfall trajectories in the a) ORG, b) DW and c) CLR states. d) Detailed three-dimensional view from above the forest road of the trajectories in b) within the deadwood section: lying deadwood in white, standing trees in yellow. Source of orthophoto (a - d): Federal Office of Topography

between 31.9 % - 36.4 % lower as the results of the experiments. This relative underestimation was for the average rotational velocities per run slightly (**ORG**, 30 %) to considerably (**DW** 15.6 %) lower. Within the simulated rotational data stream, no FI could have been detected with the same thresholds.

## 4   Discussion

The important effect of lying deadwood on the rockfall dynamics of 45 kg rocks is demonstrated by the presented experimental results. Although a certain rockfall protection effect by deadwood was known in practice, it has never been investigated with such a systematic, full-scale experiment. One out of 28 released rocks (3.6%) surpassed the piled deadwood section in the upper third of the experimental **DW** slope. Compared to **ORG** and **CLR** where 81.0 % and 78.1% of the rocks passed the deadwood area, the deadwood stopping capacity due to the deadwood barrier effect within this shape class is tremendous.

Even if the GNSS-uncertainty is taken into account still 71.4 % and 73.2% of the released rocks within the **ORG** and **CLR** set-ups reached the forest below the deadwood section. This is a main difference to the findings of Bourrier et al. (2012), where with 85.7% the large amount of released rocks impacted the four felled deadwood trunks, but only 8.5 % the rocks stopped immediately. The differences in experimental design and results underlines the fact, that realistic deadwood configurations have a better protective function than deadwood with direct ground contact. The interplay between rock and deadwood log

diameter ratio plays an important role. If the protective effect of deadwood is deliberately used as a silvicultural measure, we recommend to arrange logs on top of each other behind tree stumps (as discussed by Olmedo (2015)) in order to achieve a





more favorable rock-to-deadwood diameter-ratio. The presented study site features an acceleration zone of 37° similar to the mean slope of Bourrier et al. (2012), but flattens out in the deadwood area (33°). This could factor in to the observed higher stopping capacity of the deadwood in this study.

Although run out estimations with the shadow angle method have lost importance, since three dimensional rockfall simulations prevail, it is shown that this parameter is increased by the deadwood. As the amount of the absolute angle increase is strongly depended on the location of the deadwood section, which was in this study in the upper third, a general shadow-angle-reduction due to deadwood is not applicable.

     According to the measured in situ data (Tab. 1), the forest states **ORG** and **CLR** with high maximal accelerations per run

also showed higher mean velocities than **DW**. These measurements underlie the assumption that higher velocities increase the probability of harder impacts. Additionally, longer MROD and run times resulted in a higher probability of hitting hard ground material, especially when rocky sections are randomly distributed across the study site and more frequent at the lowest part of the slope. This effect was not visible within the simulations, as for the whole area, one single ground parameter-set was used. This additionally explains why velocities of short trajectories (**DW**) are modeled more precisely as the velocities of

trajectories with longer MROD (**ORG** and **CLR**). Locally higher $M_e$ and lower $C_d$ values could solve this issue. However, since the calibrated data should serve to model rockfall in similar forests, it is not debatable to include details down to the single rock scale.

     These dependencies of the acceleration data underlines why the use of the gyroscopic data to detect FI is theoretically more purposeful than the use of acceleration data. The here detected maximal rotational velocities per run are slightly higher

as according to the empirically derived mass - mean rotational velocity relationship from open field experiments (Caviezel et al., 2021) expected. Surprisingly, the **DW** sensor streams feature less FI per run than the other two states. As most of the rocks stopped within the deadwood section, a higher ratio of impacts was expected. This can be partly explained, if the FI are evaluated as FI·s$^{-1}$ per rock motion duration: Even if the **ORG** state still has the highest FI·s$^{-1}$, the **CLR** state shows slightly lower values than **DW**. Considering that the acceleration phase was included in all states, but accounts for a relatively

large proportion of the short **DW**-runtime, the FI·s$^{-1}$ of **DW** is rather underestimated compared to the other states. Although without visual verification not all questions concerning the influence of rotational velocities on FI can be solved, it is also clear, that solely acceleration data perform worse: As visible in Figure 3.b) there are three impacts with $\geq 10$ m s$^{-2}$, but as during the first two impacts the rotational velocity even increased, a frontal impact on a opposing object is not likely. But during the third impact, a severe drop of the rotational velocity is visible. Compared with the measured accelerations in Figure 3.c) the

**DW**-impacts are much softer, but comparable with the impacts of **CLR** in the corresponding time range, and thus slope area. The FI of the depicted sensor streams in Figure 3 shows a reduction by more than 50% from $> 40$ rad ($> 2000$°s$^{-1}$) to $<20$ rad ($< 1000$°s$^{-1}$), while the corresponding accelerations differ with values ranging from $15.0$ m $\cdot$ s$^{-1}$ to $46.9$ m $\cdot$ s$^{-1}$ by more than 300 %. To detect FI within simulations using the same thresholds, the data-output dump step has to be set on $0.001$ s which corresponds to the 1 kHz sampling rate of the StoneNode sensors making the procedure prone to jitter effects.

The protective effect of lying deadwood can only be emulated by rockfall simulations if deadwood configurations can be added realistically into the code. Here, all three forest states, incorporating the exact GNSS position as three-dimensional cones


for standing and lying trees alike could be reconstructed as input scenario for the rockfall simulations. This strict adherence to realistic site conditions enhance the significance of the model output for hazard assessment significantly. The inclusion of the deadwood logs as three dimensional cones, resulted in simulations with a realistic deposition pattern. Apart from the depicted barrier effect of single, near-slope deadwood logs, piled trunks after wind-throw events can now be incorporated into simulations. Additionally, a tunnel effect can be modeled: rocks can slip under deadwood, which is essential or if the deadwood branches are still fresh and support the log above ground. In such cases, other methods like adapting the slope in the corresponding grid cells, overestimate the deadwood effect and experience drawbacks (Fuhr et al., 2015; Costa et al., 2021).

Nevertheless, the simulations do not feature the observed experimental lateral spread. A possible explanation lies in the post-experimental timing of the LiDAR flight: the deadwood clearing work may have changed the local topography slightly, which was during the **ORG** and **DW** experiments partially responsible for the lateral spread, since the $CLR_{FIT}$ generally fits best for all soil parameters-sets.

The overestimation of the simulated run times may be caused by a major issue with any numeric rockfall code: the stopping criterion. To examine the contribution to this overestimation, we calculated the mean velocity on the last 2 m travelling distance, as the rocks could theoretically move with a very low velocity slightly above the stopping criterion (= 0.5 m·s$^{-1}$) for several seconds if the stopping criterion is set too low. As $ORG_{v-stop} = 1.5 \pm 0.4$ and $CLR_{v-stop} = 1.3 \pm 0.4$ m·s$^{-1}$ ≥ 0.5 m·s$^{-1}$, the stopping criteria can not serve as main explanation of the diverging run times. However, since the run time determines the mean rock velocity, which affects the most crucial output variable of rockfall simulation programs, the kinetic energy, a velocity estimation must be part of future experiments and taken into account for calibration.

Four main limitations came to our attention during the study, which leads to new practically relevant follow-up research questions and hypotheses, and should be included in future experimental campaigns:

1. The investigated rock mass of 45 kg represents in many stratigraphic units a common rockfall release volume with a high occurrence probability and small return period. Nevertheless, with energies ≤ 3 kJ only low rockfall intensities are achieved (FOEN, 2016). The verification of the rockfall stopping capacity under higher energies remains to be investigated.

2. For an in depth rockfall-model-calibration in forests, such a deposition pattern in different forests states is unique and allowed to examine the performance of the presented three dimensional deadwood in rockfall simulations. The in situ measurement data available beyond the deposition pattern fostered helps for a complete calibration. However, the evaluations showed that (for example) a visual check of the impact location at high acceleration measurement values is desirable. Although challenging due to the visibility restrictions because of the trees Bourrier et al. (2012), a slope-wide reconstruction of the rock velocities would complete the set of the parameters of interest (Caviezel et al., 2019a). Developments towards automated detection of tree impacts of in situ sensor data, visually coverage is necessary to gain a large enough amount of data.

3. Only compact rock shapes (Sneed and Folk, 1958) were used in this study. The results of open land experiments emphasize the importance of different rock shapes (Caviezel et al., 2021). Investigation on rockfall simulations in forests





incorporating different rock shapes (Lu et al., 2020) show the importance of real scale experiments as calibration basis as different contact behavior and MROD are expected.

4. The fresh deadwood used here originates from trees felled on site. However, for forestry practice, it is also relevant to know how decomposed deadwood protects against rockfall, how this effect is changing over time. It is essential to understand the potential adverse long-term effects of rocks deposited temporarily behind decaying deadwood as they may act as secondary rockfall sources.

Despite these still existing limitations and open questions, our results indicate that the complete removal of lying deadwood after natural disturbances or logging operations can lead to a substantial decrease in rockfall protection. While our experiments allow also to quantify such effects, more experiments and long-term studies are needed to fully quantify them for different settings and to optimise their implementation in rockfall simulation models and management guidelines.

## 5  Conclusions

This experimental rockfall trilogy within different forest states, highlights the high protection capacity of partly piled deadwood against low-energy rockfalls. This is of general interest, as natural disturbances with piled deadwood are suggested to increase in future. The ratio between the rock diameter and the overall deadwood height has a decisive impact on the rock stopping capacity, as the comparison to other studies with a lower ratio imply. The agreement achieved between the simulations and the experiments is particularly convincing for the mean runout distance, while it somewhat underestimates the lateral dispersion of the deposition points. The presented three-dimensional deadwood logs within the simulations performed realistic which affirms the demand for the inclusion of deadwood in rockfall simulations. This will allow forest managers to base their future dead wood management after natural disturbances, thinning, sanitary felling and regeneration cuts on a larger scientific basis. Based on our experimental results, we recommend at least in the case of relatively small expected rock sizes to consider supplementing the natural protection of the ecological infrastructure with additional transverse, lying deadwood logs as a cost-effective, economical, ecological, nature based protection measure. Thanks to the extended model outcomes presented here, the benefits of these measures can be evaluated and thus planned in a more systematic manner, which in turn could reduce the overall economic costs. Future studies should focus on higher rockfall energies, where the rock velocities are completely retrievable, the influence of the rock shape is in depth examined and long-term effects after partial decay of deadwood in post-disturbance stands is taken into account.

*Data availability.* The experimental deposition points, the in situ StoneNode data and the FI-detection matlab script as well as the used input data for the rockfall simulation (DEM, tree file, release point and rts. files of the trees) will be public available under doi:10.16904/envidat.248 upon publication acceptance.





*Author contributions.* AR performed the data analysis, conceived the simulations and wrote the manuscript based on discussions and improvements from all authors, ES, AC, AR carried out the model calibration, AC conceived the experiment. AC, YB, KS, SD, AS carried out the experiments. KS, MK and DM conducted the forest inventory. YB and AS operated the GNSS and GL and MC programmed the deadwood Module.

*Competing interests.* The authors declare that they have no conflict of interest.

*Acknowledgements.* We thank the forest owner: the municipality of Albula/Alvra, Switzerland, and the AWN, Region 4, Tiefencastel for the permission to conduct experiments on the Surava site. The research was partially funded by the National Research Program "Sustainable Economy: resource-friendly, future-oriented, innovative" (NRP 73) by the Swiss National Science Foundation (Grant number: 407340_172415) and is part of the WSL research program Climate Change Impacts on Alpine Mass Movements (CCAMM).



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
