# Peer review of "Full scale experiments to examine the role of deadwood on rockfall dynamics in forests"

_Natural Hazards and Earth System Sciences, 2021_

## Author Comment (AC1)

**Author's response to the Interactive comment of Franck Bourrier, Referee #1, on "Full scale experiments to examine the role of deadwood on rockfall dynamics in forests"**

Dear Franck,

We thank you for the in general positive conception of our submission and your suggestions for improvements. We amended the manuscript with respect to your advice and enriched the abstract and conclusion with further information about the rock dimensions and their effects on the non-broken deadwood logs and living trees.

Please find below the response to your remaining criticisms. A manuscript including the open points would be ready but can only be submitted after the editorial decision has been made.

**1) p.2 l. 44, 45: I don't understand the sentence.**

We rephrased the sentence to clarify the introduction of deadwood logs into the three-dimensional rockfall model.

**2) p.3 l. 63: the reference is strange**

We amended the reference.

**3) p.3 l.71-73: how were the SDE, SDEla and SDEsa calculated?**

The SDE, $SDE_{la}$ and $SDE_{sa}$ were calculated using the standard python matplotlib library, including Carsten Schelps' function "plot_confidence_ellipse.py". The two-dimensional calculation is based on a particular case to obtain the eigenvalues of the 2D dataset. The original gist.hub repository is now linked as a source in the manuscript.

**4) p.3 l.71-73: given that the distribution of the deposit is not gaussian, can you justify the use of the SDE indicators instead of "limits of a given percentile of the distribution of the deposited blocks in 2D (X,Y)" ?**

Thank you for raising this interesting question. The null hypothesis of the normal-distribution test (scipy.stats.normaltest) for the x- and y- deposition coordinates of the barely disturbed ORG and CLR states (after the rotation of the principal components) cannot be rejected (alpha =0.05). Therefore, from a statistical point of view, we cannot say the deposition pattern does not correspond to the normal distribution, which permits the use of the SDE. Solely the longitudinal component of the DW-set up follows a non-normal distribution, mainly due to the outlier, stopped at the release and the one which surpassed the DW area.

However, choosing a given percentile of the distribution of the deposited blocks in 2D could be a viable option. As stated in the manuscript, experimental boundary conditions led to fewer deposition points in the deadwood state. Inclusion or exclusion of single deposition points heavily alters the percentile number, and single outlies would be unduly weighted. Due to this concern and – as stated above – only one axis could benefit from a refined statistical analysis, we opted for a normal distribution.

We enhanced the manuscript with explanatory sentences about the premises and results of the statistical analysis.

**5) p.3 l.81-82: I don't understand the sentence.**

We rephrased the sentence with descriptive details about the used data analysis methodology.

**6) Table 1: It can be interesting to discuss more the discrepancies between the experiments and simulations in terms of velocities for ORG and CLR and rotational velocities**

We feel that the presentation and discussion of the translational velocities regarding the stopping criterion are already present in the submitted manuscript. However, we enriched and clarified the discussion about the translational velocities and enlarged the discussion of the rotation velocities. It emerges that, in particular, the *maximum* rotation velocities show a greater variability between the individual runs, as the *mean* rotation velocities. We conclude that local soil and tree conditions are responsible for this. Since such local disparities are not represented to this level of detail in the simulations (partly to prevent overfitting), the simulated maximum rotation velocities are also lower.

**7) p.6 l.123: " the surpass the DW": missing word?**

We adjusted the attached manuscript: ORG and CLR surpass the DW state.

**8) p.7 l.125 -126 : Could you present more details about these results (quantitative comparisons between simulations for different values of the parameters, for example) ?**

In order to be able to compare the discrepancy between the input soil parameter pairs, we enhanced the results section by inserting the values $OF_{3000}$ of the 10[th] placed parameter pair. Additionally, we discussed the meaning of the observed variabilities in the discussion.

**9) p.8 l. 146 – 149 : in my opinion, the differences with the results from Bourrier et al . (2012) are mainly due to the sizes of the blocks used in the simulations. In Bourrier et al., the blocks were large enough to break the trees which completely changes the processes as well as the efficacy of the protective measure.**

Unfortunately, nothing about tree breakage was stated in Bourrier (2012), therefore we concluded that the lower stopping capacity of the deadwood was mainly due to rolling over and overjumping the stems with diameters in the same range as the rock diameters. But we enhanced the manuscript with the sentence: "The ratio between rock and deadwood log diameter plays an important role, both in terms of hindering rolling over of the obstacle as well as in terms of breaking prevention ".

Nevertheless, this underlines the importance of stating the apparent (low rockfall energies, no tree, nor deadwood breakage) in abstract and conclusion, as you proposed in your overall review.

**10) p.8 l. 151 : "Olmedo (2015)" - it can be useful to cite also "Olmedo, I., Bourrier, F., Bertrand, D., Berger, F., Limam, A. Dynamic analysis of wooden rockfall protection structures subjected to impact loading using a discrete element model**

*(2020) European Journal of Environmental and Civil Engineering, 24 (9), pp. 1430-1449."*

We have expanded our bibliography with the here suitable and suggested literature.

*11) p.9 l. 168 : "underline" instead of "underlines" ?*

We corrected the verb conjugation.

*12) p.9 l. 169: "The here" - missing word?*

We resolved this issue due to the rephrasing of the entire paragraph (see your item 13).

*13) p.9 l. 168-184: this section is not clear: it can be improved,*

We have amended the relevant section and clarified our statement about the absolute and runtime-relative number of frontal impacts per forest state. The section has been enriched with additional content (see explanations to item 6)

*14) p.9 l. 182: "m. s-2" instead of "m.s-1"*

We adjusted the wrong units in the attached manuscript.

*15) p.10 l. 202-204: I don't understand*

We enhanced this section and clarified our statement about the stopping criterion. As mentioned in item 6) the discussion about the associated disparities between the mean translational velocities of the simulation and experiments, is added to this paragraph.

*16) p.10 l. 215 : "trees Bourrier et al. (2012)" : typo ?*

We adjusted the typo within the source directly in the attached manuscript.

---

## Author Comment (AC2)

**Author's response to the Interactive comment of Anonymous Referee #2, on "Full scale experiments to examine the role of deadwood on rockfall dynamics in forests"**

Dear Anonymous Referee #2,

Thank you for highlighting the relevance of the manuscript's topic and, in agreement with most of Referee#1's comments, for the overall positive assessment of the paper. In compliance with your report, we amended the manuscript to stress the small rock masses in both abstract and conclusion. Also, the assumptions about rock-log diameter ratios were elaborated in more detail.

We willingly incorporate the suggestion of *logging operation* instead of *logging job*. We noticed that no recommendation for improvement was given for *forest state*, also criticized at the beginning. Although not completely satisfied with the term initially, as hardly ever used in the forest community, we replace it with *the state of the forest*: *The state of the forest* implies a short, defined observation period of an existing forest. If solely simulations – without actual experiments - were carried out, we would use *scenario*. And the term *forest stage*, used in ecological vocabulary, describes per se longer periods during which natural processes are responsible for changes and therefore excludes it from further considerations. The closest to the used *the state of the forest* is *forest condition*. However, the literature with *forest condition* in the title deals also with relatively long-lasting changes and their monitoring (e.g., air pollution). Although *the state of the forest* may be a bit wordier than the initially used *forest state*, it is universal and sounds less pompous.

See below the other point-by-point responses to your remaining criticisms:

*1) Line 13. "rockfall energies": use magnitude or refer to kinetic energy of a single rock*
We replaced the ambiguous term "rockfall energies" with "rockfall magnitudes".

*2) Line 32. Use uprooted trees instead of overturned trees*
We substitute "uprooted" for "overturned" trees to use more common forestry terms.

*3) Line 36. You are not referring to protection "effect" here, but to protective function or role. In order to clarify the terms to be used all over the manuscript: protection forest are forest with a protective role/function, even if they are not providing protection (not protective)*
We agree, that we were referring to the protection "role" and not the protection effect in this context and changed the wording in the manuscript to: "In rockfall protection forests, it is decisive to know. "

*4) Line 40-43. Delete the part of the sentence related to the wood decay since is not applying to this work*
We have deleted the part of the sentence at the respective position.

*5) Line51-54. The species should be written not with the first letter uppercase in vernacular (rowan, whitebeam, beech, mountain pine, silver fir) unless it is a Country name (Norway spruce, European larch). Furthermore, for the scientific name you have to provide the authority name to all the species, not only for Norway spruce and rowan.*
We integrated all the suggested adaptions, amended the taxonomic ranking with correct order names in the manuscript.

**6) Line 56. Logging operations**

We changed the term logging job at the given position into logging operation and specified it further as a regeneration cut.

**7) Line 65-67. This details are not needed as well as line 68-69**

The presence of this information was to give insights into experimental boundary conditions, which - we agree – are also a matter of taste if necessary in a manuscript. We approve deleting the passage, as the run numbers are also mentioned in Table 1. We restrained from deleting the horizontal accuracy of the GNSS device, since we refer to this matter in the results section: *"Even if the GNSS-uncertainty is taken into account still 71.4% and 73.2% of the released rocks within the ORG and CLR set-ups reached the forest below the deadwood section."* and want to highlight that factory accuracies hardly ever apply in forested test sites as information for the general, open-land GNSS user.

**8) Line 67. Delete "in" after measured**

The mistranslated Germanism was corrected.

**9) Line 85-86. Please rephrase the sentence**

We clarified the statement and rephrased the sentence.

**10) Line 88-89. Not clear to me "we took for all deadwood-logs a maximal diameter of 40 cm"**

We enhanced the mentioned passage with a statement about the missing diameter- and height above ground information of each log end. With this in mind, and further clairfications the text is now clear:

*During fieldwork, all 26 trunk GNSS positions were recorded, but not the exact diameters nor the height above ground of every log end, which are required as input parameters for the generation of the individual deadwood cone in RAMMS::ROCKFALL. As a realistic but simplified approach, we assumed a uniform maximal diameter of 40 cm for all deadwood-logs. If logs were lying on top of each other, we considered the in the GNSS-file first mentioned as the lower log with ground contact, and the latter piling up.*

**11) Line 137. Slightly and considerably values are switched?**

The statement has been rephrased entirely for clarity:

*A less pronounced underestimation, 15.6% for DW, 30% for ORG, was detected for the average rotational velocities.*

**12) Line 144. Avoid using "tremendous".....**

We replaced tremendous with substantial.

**13) Line 146-148. Please rephrase**

We have improved the incorrect expressions in the sentence

**14) Line 233. Rephrase the concept "natural disturbances with piled deadwood"**

We adjusted the sentence to indicate that piled deadwood results from the increasing natural disturbances.

---

## Author Response (AR1)

**Author's response to the Interactive comment of Franck Bourrier, Referee #1, on "Full scale experiments to examine the role of deadwood on rockfall dynamics in forests"**

Dear Franck,

We thank you for the in general positive conception of our submission and your suggestions for improvements. We amended the manuscript with respect to your advice and enriched the abstract (p. 1, l. 4 and l. 11-12) and conclusion (p. 12, l. 260, l. 265 and l. 270) with further information about the rock dimensions and their effects on the non-broken deadwood logs and living trees.

Please find below and in the attached manuscript the response to your remaining criticisms:

*1) p.2 l. 44, 45: I don't understand the sentence.*

We rephrased the sentence to clarify the introduction of deadwood logs into the three-dimensional rockfall model (see highlighted lines in the manuscript: p. 2, l. 45-47)

*2) p.3 l. 63: the reference is strange*

We amended the reference.

*3) p.3 l.71-73: how were the SDE, SDEla and SDEsa calculated?*

The SDE, $SDE_{la}$ and $SDE_{sa}$ were calculated using the standard python matplotlib library, including Carsten Schelps' function "plot_confidence_ellipse.py". The two-dimensional calculation is based on a particular case to obtain the eigenvalues of the 2D dataset. The original gist.hub repository is now linked as a source in the manuscript (p. 3, l. 74).

*4) p.3 l.71-73: given that the distribution of the deposit is not gaussian, can you justify the use of the SDE indicators instead of "limits of a given percentile of the distribution of the deposited blocks in 2D (X,Y)" ?*

Thank you for raising this interesting question. The null hypothesis of the normal-distribution test (scipy.stats.normaltest) for the x- and y- deposition coordinates of the barely disturbed ORG and CLR states (after the rotation of the principal components) cannot be rejected (alpha =0.05). Therefore, from a statistical point of view, we cannot say the deposition pattern does not correspond to the normal distribution, which permits the use of the SDE. Solely the longitudinal component of the DW-set up follows a non-normal distribution, mainly due to the outlier, stopped at the release and the one which surpassed the DW area.

However, choosing a given percentile of the distribution of the deposited blocks in 2D could be a viable option. As stated in the manuscript, experimental boundary conditions led to fewer deposition points in the deadwood state. Inclusion or exclusion of single deposition points heavily alters the percentile number, and single outlies would be unduly weighted. Due to this concern and – as stated above – only one axis could benefit from a refined statistical analysis, we opted for a normal distribution.

We enhanced the manuscript with explanatory sentences about the premises and results of the statistical analysis (p. 5, l. 110 and p. 6, l. 122).

*5) p.3 l.81-82: I don't understand the sentence.*

We rephrased the sentence with descriptive details about the used data analysis methodology (p. 3, l. 80-85). For further clarifications, we enriched Figure 3 with content, highlighting the frontal impact flagging procedure, further clarifying the presented work.

*6) Table 1 : It can be interesting to discuss more the discrepancies between the experiments and simulations in terms of velocities for ORG and*
*CLR and rotational velocities*

We feel that the presentation and discussion of the translational velocities regarding the stopping criterion are already present in the submitted manuscript (p. 10, l. 220 ff. in the old manuscript). However, we enriched and clarified the discussion about the translational velocities (p.10, l. 222 – 232) and enlarged the discussion of the rotation velocities (p. 9, l.184). It emerges that, in particular, the *maximum* rotation velocities show a greater variability between the individual runs, as the *mean* rotation velocities. We conclude that local soil and tree conditions are responsible for this. Since such local disparities are not represented to this level of detail in the simulations (partly to prevent overfitting, see p. 9, l. 180-182), the simulated maximum rotation velocities are also lower.

*7) p.6 l.123: " the surpass the DW": missing word?*

We adjusted the attached manuscript: ORG and CLR surpass the DW state (p. 6, l. 135)

*8) p.7 l.125 -126 : Could you present more details about these results (quantitative comparisons between simulations for different values of the parameters, for example) ?*

In order to be able to compare the discrepancy between the input soil parameter pairs, we enhanced the results section by inserting the values $OF_{3000}$ of the 10th placed parameter pair (p. 6, l. 138 – 140).

*9) p.8 l. 146 – 149 : in my opinion, the differences with the results from Bourrier et al . (2012) are mainly due to the sizes of the blocks used in the simulations. In Bourrier et al., the blocks were large enough to break the trees which completely changes the processes as well as the efficacy of the protective measure.*

Unfortunately, nothing about tree breakage was stated in Bourrier (2012), therefore we concluded that the lower stopping capacity of the deadwood was mainly due to rolling over and overjumping the stems with diameters in the same range as the rock diameters. But we enhanced the manuscript with the sentence: "The ratio between rock and deadwood log diameter might play an important role, both in terms of hindering rolling over of the obstacle as well as in terms of breaking revention, as the reported ratio in Bourrier et al. (2012) < 1 and the here observed ratio > 1 imply. " (p. 9, l. 163 – 165).
Nevertheless, this underlines the importance of stating the apparent (low rockfall energies, no tree, nor deadwood breakage) in abstract and conclusion, as you proposed in your overall review.

*10) p.8 l. 151 : "Olmedo (2015)" - it can be useful to cite also "Olmedo, I., Bourrier, F., Bertrand, D., Berger, F., Limam, A. Dynamic analysis of wooden rockfall protection structures subjected to impact loading using a discrete element model*

*(2020) European Journal of Environmental and Civil Engineering, 24 (9), pp. 1430-1449."*

We have expanded our bibliography with the here suitable and suggested literature (p. 9, l. 167).

*11) p.9 l. 168 : "underline" instead of "underlines" ?*

We corrected the verb conjugation (p.9, l.175)

*12) p.9 l. 169: "The here" - missing word?*

We resolved this issue due to the rephrasing of the entire paragraph (see your item 13).

*13) p.9 l. 168-184: this section is not clear: it can be improved,*

We have amended the relevant section and clarified our statement about the absolute and runtime-relative number of frontal impacts per forest state (p. 9, l. 183-217). The section has been rearranged and enriched with additional content (see explanations to item 6).

*14) p.9 l. 182: "m. s-2" instead of "m.s-1"*

We adjusted the wrong units in the attached manuscript (p. 10, l. 206)

*15) p.10 l. 202-204: I don't understand*

We enhanced this section and clarified our statement about the stopping criterion (p. 10, l. 222-232). As mentioned in item 6) the discussion about the associated disparities between the mean translational velocities of the simulation and experiments, is added to this paragraph.

*16) p.10 l. 215 : "trees Bourrier et al. (2012)" : typo ?*

We adjusted the typo within the source directly in the attached manuscript (p. 11, l. 243).

**Author's response to the Interactive comment of Anonymous Referee #2, on "Full scale experiments to examine the role of deadwood on rockfall dynamics in forests"**

Dear Anonymous Referee #2,

Thank you for highlighting the relevance of the manuscript's topic and, in agreement with most of Referee#1's comments, for the overall positive assessment of the paper. In compliance with your report, we amended the manuscript to stress the small rock masses in both abstract (p. 1, l. 4) and conclusion (p. 12, l. 260). Also, the assumptions about rock-log diameter ratios were elaborated in more detail (p. 9, l. 163- 165) and the conclusion adapted.

We gladly incorporate the suggestion of *logging operation* instead of *logging job*. We noticed that no recommendation for improvement was given for *forest state*, also criticized at the beginning. Also not completely satisfied with the term initially, as hardly ever used in the forest community, we replace it solely with *the state of the forest*: *The state of the forest* implies a short, defined observation period of an existing forest. If solely simulations – without actual experiments - were carried out, we would use *scenario*. And the term *forest stage*, used in ecological vocabulary, describes per se longer periods during which natural processes are responsible for changes and we therefore excluded it from further considerations. The closest to the used *the state of the forest* is *forest condition*. However, the literature with *forest condition* in the title deals also with relatively long-lasting changes and their monitoring (e.g., air pollution). Although *the state of the forest* may be a bit wordier than the initially used *forest state*, it is universal and sounds less pompous.

See below the other point-by-point responses to your remaining criticisms:

*1) Line 13. "rockfall energies": use magnitude or refer to kinetic energy of a single rock*
We replaced the ambiguous term "rockfall energies" with "rockfall magnitudes" (p. 1, l. 15).

*2) Line 32. Use uprooted trees instead of overturned trees*
We substitute "uprooted" for "overturned" trees to use more common forestry terms (p. 2, l. 34).

*3) Line 36. You are not referring to protection "effect" here, but to protective function or role. In order to clarify the terms to be used all over the manuscript: protection forest are forest with a protective role/function, even if they are not providing protection (not protective)*
We agree, that we were referring to the protection "role" and not the protection effect in this context and changed the wording in the manuscript to: "In rockfall protection forests, it is decisive to know." (p. 2, l. 38).

*4) Line 40-43. Delete the part of the sentence related to the wood decay since is not applying to this work*
We have deleted the part of the sentence at the respective position (p. 2, l. 43).

*5) Line51-54. The species should be written not with the first letter uppercase in vernacular (rowan, whitebeam, beech, mountain pine, silver fir) unless it is a Country name (Norway spruce, European larch). Furthermore, for the scientific name you have to provide the authority name to all the species, not only for Norway spruce and rowan.*
We integrated all the suggested adaptions, amended the taxonomic ranking with correct order names in the manuscript (p. 3, l. 54 - 56).

**6) Line 56. Logging operations**

We changed the term logging job at the given position into logging operation and specified it further as a regeneration cut (p. 3, l. 59).

**7) Line 65-67. This details are not needed as well as line 68-69**

The presence of this information was to give insights into experimental boundary conditions, which - we agree – are also a matter of taste if necessary in a manuscript. We approve deleting the passage, as the run numbers are also mentioned in Table 1. We restrained from deleting the horizontal accuracy of the GNSS device (p. 3, l. 69 – 70), since we refer to this matter in the results section: *"Even if the GNSS-uncertainty is taken into account still 71.4% and 73.2% of the released rocks within the ORG and CLR set-ups reached the forest below the deadwood section." (p. 8-9, l. 158 – 160)* and want to highlight that factory accuracies hardly ever apply in forested test sites as information for the general, open-land GNSS user.

**8) Line 67. Delete "in" after measured**

The mistranslated Germanism was corrected (p. 3, l. 69).

**9) Line 85-86. Please rephrase the sentence**

We clarified the statement and rephrased the sentence (p. 4, l. 87 – 90).

**10) Line 88-89. Not clear to me "we took for all deadwood-logs a maximal diameter of 40 cm"**

We enhanced the mentioned passage with a statement about the missing diameter- and height above ground information of each log end. With this in mind, and further clarifications the text is now clear:

*During fieldwork, all 26 trunk GNSS positions were recorded, but not the exact diameters nor the height above ground of every log end, which are required as input parameters for the generation of the individual deadwood cone in RAMMS::ROCKFALL. As a realistic but simplified approach, we assumed a uniform maximal diameter of 40 cm for all deadwood-logs. If logs were lying on top of each other, we considered the in the GNSS-file first mentioned as the lower log with ground contact, and the latter piling up.* (p. 4, l. 90 - 94)

**11) Line 137. Slightly and considerably values are switched?**

The statement has been rephrased entirely for clarity:
*A less pronounced underestimation, 15.6% for DW, 30% for ORG, was detected for the average rotational velocities.* (p. 7, l. 150 – 151)

**12) Line 144. Avoid using "tremendous".....**

We replaced tremendous with substantial (p. 8, l. 158).

**13) Line 146-148. Please rephrase**

We have improved the incorrect expressions in the sentence (p. 9, l. 168-169).

**14) Line 233. Rephrase the concept "natural disturbances with piled deadwood"**

We adjusted the sentence to indicate that piled deadwood results from the increasing number of natural disturbances (p. 12, l. 263).

---

## Author Response (AR2)

**Author's response to the comment of executive editor Paolo Tarollli and Anonymous referee #2 on "Full scale experiments to examine the role of deadwood on rockfall dynamics in forests"**

**Dear Executive Editor Paolo Tarolli,**

Thank you for your positive feedback. We resolved the raised technical corrections as per the below point-to-point reply to Anonymous referee #2.

**Dear Anonymous referee #2**

Thank you for your invested time in evaluating our manuscript and the in general positive review. A native-speaking co-author revised the manuscript again and corrected several linguistic inconsistencies throughout the whole manuscript: Additionally, we tackled all your mentioned points. See the changes written in blue in the below-attached manuscript

Line 45-47 - this sentence should be enhanced  $\rightarrow$  DONE Line 59-60 - improve the sentence, remove former. The intervention was the creation of a gap (small size clearcut) in the upper part of an even-aged spruce forest.  $\rightarrow$  DONE Line 70 - It is better to use "north-west-facing slope"  $\rightarrow$  DONE Line 88 & 93 - the in, the in the  $\rightarrow$  DONE Line 168-170 - rephrase and explain better  $\rightarrow$  DONE Line 186 - forest states, not forestall  $\rightarrow$  DONE

**Full scale experiments to examine the role of deadwood on rockfall dynamics in forests**

Adrian Ringenbach1,2,4, Elia Stihl1,2, Yves Bühler1,2, Peter Bebi1,2, Perry Bartelt1,2, Andreas Rigling3,4, Marc Christen1,2, Guang Lu1, Andreas Stoffel1,2, Martin Kistler1, Sandro Degonda1, Kevin Simmler1, Daniel Mader3, and Andrin Caviezel1,2

[revised manuscript text omitted]